# Continual Learning via Sparse Memory Finetuning

## Abstract

Modern language models are powerful, but typically static after deployment. A major obstacle to building models that continually learn over time is catastrophic forgetting, where updating on new data erases previously acquired capabilities. Motivated by the intuition that mitigating forgetting is challenging because trainable parameters are shared across all tasks, we investigate whether *sparse parameter updates* can enable learning without catastrophic forgetting. We introduce sparse memory finetuning, leveraging memory layer models (Berges et al., 2024), which are sparsely updated by design. By updating only the memory slots that are highly activated by a new piece of knowledge relative to usage on pretraining data, we reduce interference between new knowledge and the model's existing capabilities. We evaluate learning and forgetting compared to full finetuning and parameter-efficient finetuning with LoRA on two question answering tasks. We find that sparse memory finetuning learns new knowledge while exhibiting substantially less forgetting: while NaturalQuestions F1 drops by 89% after full finetuning on new facts and 71% with LoRA, sparse memory finetuning yields only an 11% drop with the same level of new knowledge acquisition. Our results suggest sparsity in memory layers offers a promising path toward continual learning in large language models.

## 1 Introduction

Language models store vast amounts of information in their parameters, but this knowledge largely remains fixed after pretraining. How can we build systems that can continually accumulate knowledge through experience and interaction with the world? This capability for *continual learning* would enable models that remember what we teach them, learn from mistakes, and acquire new skills in the real world.

Continual learning has been a longstanding challenge in AI, and persists in the era of large language models (LLMs). A key barrier to continual learning is *catastrophic forgetting* (McCloskey & Cohen, 1989): when updating on a stream of new information, models often lose previously acquired capabilities. While replaying data from pre-training or previous training stages can alleviate forgetting (Lin, 1992; Robins, 1995; Chen et al., 2025), this strategy is data-inefficient and not scalable as we grow the amount of experience. Already, modern LLMs undergo multiple rounds of pre-training, post-training, and alignment, making replay an increasingly difficult and delicate strategy to implement. A fundamental problem is that we optimize the same set of parameters across the lifetime of a model, leading to interference unless parameters are optimized simultaneously for all downstream tasks.

In this work, we propose a new approach to continual updates, enabled by memory layers (Berges et al., 2024; He, 2024). Our key observation is that memory layers access a small set of parameters (e.g., 10k) out of a large memory pool (1-10M) on each forward pass, striking a balance between large overall capacity and a minimal set of parameters for each piece of knowledge. We introduce *sparse memory finetuning*, updating just the top $t$ memory slots that are more frequently accessed on a certain batch relative to some background corpus (e.g. pretraining data). We use TF-IDF as a ranking score, identifying a set of indices to update with each gradient step that minimally interferes with the model's existing knowledge.

We evaluate the learning and forgetting behavior of sparse memory finetuning on two tasks: (1) learning from a stream of facts and (2) learning from a stream of documents, without degrading capabilities from pretraining. We show that sparse memory finetuning learns new knowledge to the same degree as full finetuning, but with minimal to no degradation on held-out benchmarks. In contrast, full finetuning and parameter-efficient finetuning with LoRA (Hu et al., 2021), exhibit catastrophic forgetting even within a thousand gradient steps. When training on a stream of TriviaQA facts, performance on NaturalQuestions drops by 89% with full finetuning and 71% with LoRA, but only 11% with sparse memory finetuning with the same level of retention. Our results suggest that sparsity is potentially a key ingredient for continual learning, with sparse finetuning of memory layers as a promising implementation.

## 2 RELATED WORK

A longstanding goal in AI is continual or lifelong learning: the ability to accumulate new knowledge and skills over time. A major challenge is catastrophic forgetting, where incorporating new knowledge into a neural network results in loss of previously acquired knowledge or catastrophic forgetting (McCloskey & Cohen, 1989; French, 1999), which persists with modern LLMs (Luo et al., 2023; Lai et al., 2025).

A variety of strategies have been proposed to address this problem. Regularization methods such as dropout (Srivastava et al., 2014), weight decay (Loshchilov & Hutter, 2019), or KL penalties (Ouyang et al., 2022) restrict parameter updates to preserve performance to stay close to initialization. Elastic Weight Consolidation (Kirkpatrick et al., 2017) regularizes updates to preserve parameters that are "important" to previous tasks, as measured with the Fisher information matrix. Our work leverages a similar idea, but implements it by selecting parameters in a memory layer, which enables fully sparsified updates, rather than regularization penalties. Expansion-based approaches add new parameters such as adapter, LoRA modules, or MoE experts for each task (Rusu et al., 2016; Wang et al., 2024; Houlsby et al., 2019; Hu et al., 2021; Gritsch et al., 2024; Shen et al., 2023). LoRA has become a popular method for adding lightweight task-specific parameters, but Biderman et al. (2024) show that while LoRA achieves less forgetting, it learns less. Parameter-efficient expansion fundamentally add only a small amount of capacity, limiting the amount of new knowledge they can support. In contrast, our approach relies on memory layers, which strike a balance between learning capacity and forgetting by using a sparsely indexed memory pool with a large overall capacity (1B+ parameters). Finally, replay-based methods reduce forgetting by maintaining a buffer of previous tasks or pretraining samples to rehearse during training (Robins, 1995; Lesort et al., 2022; Scialom et al., 2022; Chen et al., 2025). While popular and effective, replay is data-inefficient: as models gain more experience, they must rehearse ever larger corpora. Modern LLMs also go through many rounds of training, adding significant complexity when we want to e.g. preserve the model's pretraining knowledge while also maintaining its instruction-following capabilities.

Our method leverages sparsity as an alternative approach to continual learning, with the key intuition that only a small percentage of the model's parameters truly need to be updated on each input. Work such as grafting has found that as little as 0.01% of model parameters are responsible for model performance on a particular task (Panigrahi et al., 2023), and that these parameters can be isolated to enable continual learning with less forgetting. In our work, we use recently-proposed memory layers (Berges et al., 2024; He, 2024) that are sparsely indexed by design to continuously update the model with new knowledge without interfering with parameters for other knowledge.

## 3 BACKGROUND

Memory layers (Berges et al., 2024; He, 2024; Weston et al., 2015) add a trainable parametric memory that can be queried via an attention-like mechanism. We illustrate this in Figure 1. The standard Transformer block consists of a self-attention layer, followed by a feedforward network. To augment the model with memory, some of the feedforward layers can be replaced with a memory lookup into a pool of memory of size $N$, with keys $K \in \mathbb{R}^{N \times d}$ and values $V \in \mathbb{R}^{N \times d}$. Given an

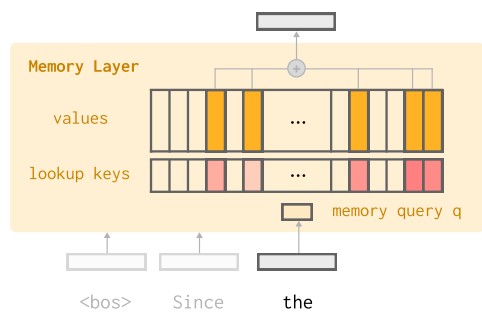

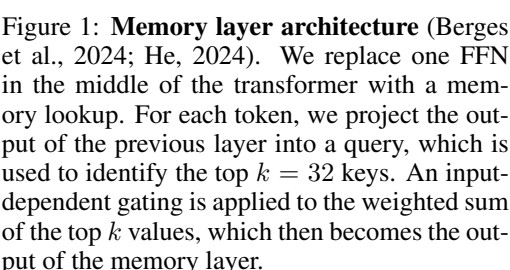

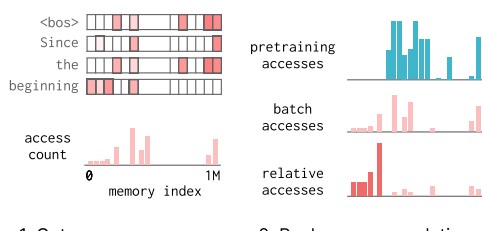

1. Get memory accesses
in batch

2. Rank accesses relative
to background corpus and
train the top t

Figure 1: **Memory layer architecture** (Berges et al., 2024; He, 2024). We replace one FFN in the middle of the transformer with a memory lookup. For each token, we project the output of the previous layer into a query, which is used to identify the top $k = 32$ keys. An input-dependent gating is applied to the weighted sum of the top $k$ values, which then becomes the output of the memory layer.

Figure 2: **Sparse memory finetuning.** To identify the minimal set of memory slots to update, we aggregate counts for all the memory indices on a particular batch, and then rank the accesses relative to a background corpus (e.g. pretraining data) with TFIDF. We train the top $t$ memory indices that are highly accessed on this batch but accessed relatively infrequently on pretraining, keeping the rest of the memory pool and model frozen.

input $x \in \mathbb{R}^n$ and query projection $q : \mathbb{R}^n \to \mathbb{R}^d$,

$$\mathbb{I} = \text{TopKIndices}(Kq(x), k) \qquad \text{\# Retrieve top-k indices}$$
$$s = \text{softmax}(K_{\mathbb{I}}q(x)) \qquad \text{\# Compute scores}$$
$$y = sV_{\mathbb{I}} \qquad \text{\# Compute weighted output}$$
$$\text{output} = (y \odot \text{silu}(x^\intercal W_1))^\intercal W_2 \qquad \text{\# Apply input-dependent gating}$$

where $K_{\mathbb{I}} \in \mathbb{R}^{k \times d}$ are the top-k keys, $W_1 \in \mathbb{R}^{n \times d}$ and $W_2 \in \mathbb{R}^{d \times n}$ are learned projection matrices, and $\text{silu}(x) = x \, \text{sigmoid}(x)$. As in attention, we can make a memory layer more expressive by adding additional heads with different key projections. As typical memory sizes are 1M-100M keys for billion-parameter base models, a bottleneck is the memory embedding lookup. To perform memory lookups efficiently, memory layers use product keys (Lample et al., 2019) to decompose the keys into two halves, enabling efficient lookup across a large number of indices.

Unlike attention, the keys and values are trainable parameters (rather than activations) that can be finetuned. This approach can also be thought of as a mixture-of-experts (MoE) architecture (Shazeer et al., 2017) with a large number of small experts, one for each memory location (He, 2024). Memory layers provide several benefits over MoEs: since each token only activates a small set of parameters rather than a large expert, decoding efficiency can be much improved, given the memory-bound nature of inference. In this work, we leverage the benefit that memory layers provide much more granular control over how information is accessed and stored in parameters: rather than activating one of (typically) 10-100 experts, each token activates only a tiny subset of the total memory parameters (e.g. on the order of 0.03%-0.0002% of total memory parameters).

## 4 SPARSE MEMORY FINETUNING

Our work proposes finetuning the model on new knowledge via sparser updates to the memory layer. The memory layer is already sparsely indexed on each forward pass, with only $k$ values out of the entire memory pool accessed at each token (e.g. $k = 32$ per memory attention head, out of 1M total indices). However, we find that naively finetuning the memory layer model still causes catastrophic forgetting (see Section 6). Some of the memory indices accessed on a particular batch may serve general purposes, such as helping predict syntactic structures or features of the broader domain. Intuitively, we'd like to finetune the minimal set of parameters that "store" a piece of knowledge.

To implement this, we propose to update only the memory slots that are *specific* to a particular input: i.e., highly accessed specifically for this input, compared to memory accesses on other inputs. This problem frequently shows up in document retrieval, e.g. to measure the importance of particular words in a document with TF-IDF by looking at how frequently they appear in this document, relative to overall occurrence in all documents. We adopt TF-IDF as the ranking metric to identify memory indices that are important on a particular *batch*, although future work can explore more sophisticated scoring functions or granularities for ranking (e.g., choosing sequence-level rather than batch-level memory indices).

For a given batch, we count all the memory accesses. We then compute TF-IDF score for each memory index relative to the indices accessed on some *background corpus* of knowledge we want to preserve (for the IDF portion of the computation). For our main experiments, we use the memory accesses on 1000 random batches of DCLM (Li et al., 2024) as a representative sample of generic pretraining data. These "background indices" do not change during finetuning and can be stored statically in the model checkpoint. We study how the choice of background corpus affects learning and forgetting in Section 6. We note that ranking based on batches makes no assumption about task boundaries; consecutive batches can be from the same or totally different data distributions.

For a given memory slot $i \in M$ (where $M$ is all memory slots), the TF-IDF score is:

$$\frac{c(i)}{\sum_{j \in M} c(j)} \cdot \log \frac{|B| + 1}{\sum_{b \in B} \mathbf{1}_{c_b(i) > 0} + 1}$$

where $c(i)$ is the number of times memory index $i$ is accessed on this batch, $c_b(i)$ is the number of times $i$ is accessed on some batch $b$, and $B$ is the set of background (DCLM) batches. Then, we finetune the values of the top $t$ memory slots. On the forward pass, all accessed indices contribute to the model output, but we stop the gradient on all memory parameters except for the top $t$ indices after TF-IDF ranking. Since the top $t$ indices is dynamic on each forward pass, this can be implemented as follows:

```
# trainable_mask: mask of shape (memory_size, 1), 1 if trainable
# mem: memory table, shape (memory_size, value_dim)

# The value of memory is unchanged, but the gradient goes through mask
mem = mem * trainable_mask + mem.detach() - (mem * trainable_mask).detach()
```

Each forward pass typically accesses $10^3$ to $10^6$ indices (k * num_memory_heads * batch_size * seqlen), but we find empirically that we can set $t$ to much lower values while achieving the same learning performance.

## 5 EXPERIMENTS

We compare sparse memory finetuning to full finetuning and parameter-efficient finetuning with LoRA (Hu et al., 2021). LoRA has been shown to mitigate forgetting (Biderman et al., 2024), making it a natural and strong baseline for our experiments. We use a 1.3B base model for our main experiments, pretrained on the same data for all methods. We use a batch size of 64 and sequence length of 64 for TriviaQA and 512 for SimpleQA. For the memory-augmented model, we swap out the feedforward network (FFN) in the middle of the model (layer 12 out of 22) with a lookup into a memory pool of size 1M, $k = 32$ memory accesses per token, 4 memory heads, and a value dimension of 1024. This amounts to $32 * 1024 = 32,768$ active parameters in this layer instead of 50M parameters in original FFN (with a model dim of 2048 and FFN dim of $2048 * 4$). For LoRA finetuning, we apply LoRA to all attention and FFN weight matrices, and report the best performing setting of rank and alpha. For full finetuning, we report the best performing learning rate.

**Optimizer.** We initially used AdamW for all methods before realizing that adaptive per-parameter step sizes, weight decay, and momentum can interact with sparsity in unexpected ways. Even controlling for the optimizer (AdamW for all methods), sparse memory finetuning achieved similar learning with less forgetting than full finetuning and LoRA. Switching to SGD further decreased the forgetting on held-out tasks, although interestingly we did not see similar benefits for full finetuning

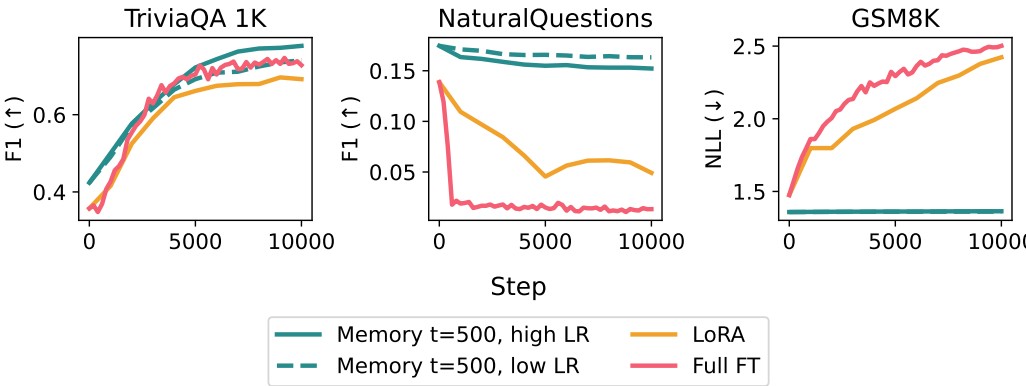

Figure 3: **Learning single facts in the small-data regime.** To simulate a "small data" regime where the model must perform immediate updates on a small amount of data, we train the model to learn a sequence of 1000 facts from TriviaQA. Sparse memory finetuning learns more on the target facts, while forgetting much less on held-out benchmarks (NaturalQuestions and GSM8K). LoRA and full finetuning exhibits catastrophic forgetting on the held-out metrics.

and LoRA (see Appendix B for baseline results with SGD). In our experiments, we use the best performing optimizer for each method (AdamW with $\lambda = 0.1$ for baselines, SGD for sparse memory finetuning). Previous work has also suggested that Adam may not be appropriate for continual learning (Hsu et al., 2019), but we leave a more thorough exploration of appropriate optimizers to future work.

### 5.1 FACT LEARNING

Many desired applications for continual learning require integrating a small amount of new information into the model (e.g. personalization or learning from feedback from individual users). Unlike continued pre-training on documents, where we can more easily augment or mix in diverse data, this setting poses a special challenge for finetuning. The amount of data to learn from is both small and narrow in domain (e.g. learning a user preference from a single message), making it more likely that continued gradient updates will lead to model collapse and forgetting of general capabilities.

To test methods in the "small-data" regime, we consider a setting where the model must learn single facts in sequence. We use 1K questions from the TriviaQA test set and rephrase them as statements. To fill the batch, we paraphrase the statement $N$ times to fill a batch size of $N$. We pad paraphrases to the max sequence length, since predicting tokens of a paraphrase is trivial if there are identical paraphrases in context, leading to different parameter updates and memory accesses. For memory finetuning, we take care to mask the indices accessed at the padding locations. We find that a top $t = 500$ is sufficient for best performance, and report results for this setting.

In Figure 3, we see that sparse memory finetuning learns more, while forgetting less on held-out benchmarks.[1] For held-out performance, we measure F1 score on NaturalQuestions (Kwiatkowski et al., 2019) and accuracy on HellaSwag (Zellers et al., 2019). We report results with both high (lr=5) and low (lr=2) learning rates for memory finetuning to characterize its behavior: lower learning rates can better preserve performance on held-out tasks, while still matching or exceeding the target performance of baselines. Memory finetuning continuously improves in performance, with significantly less degradation in held-out metrics thanks to selective updates.

### 5.2 DOCUMENT QA

For our second task, we evaluate whether models can learn from a continuous stream of documents.

---

[1]The base memory-augmented model starts at a higher performance due to better knowledge retention in pretraining, as also observed in Berges et al. (2024).

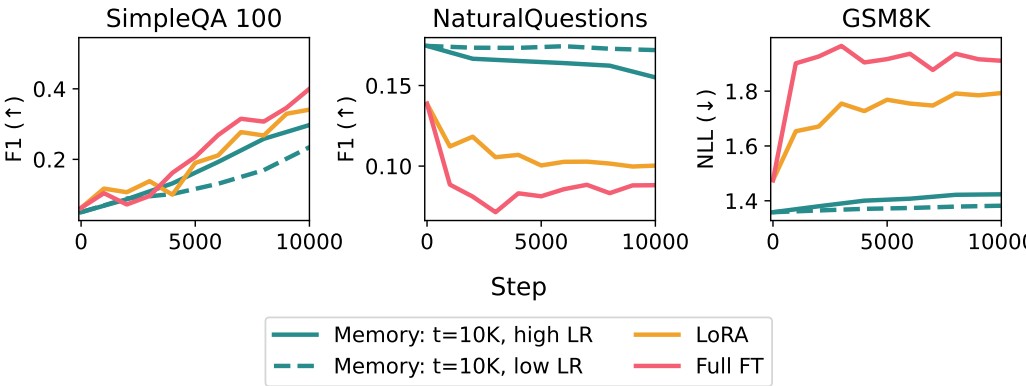

Figure 4: **Learning from documents.** We evaluate whether methods can learn from a stream of documents by training on Active Reading-augmented (Lin et al., 2025) documents for a subset of 100 Wikipedia-grounded SimpleQA questions (Wei et al., 2022). The best full finetuning and LoRA configurations achieve the same performance on the target task, but suffer from much more forgetting on held-out tasks. Sparse memory finetuning achieves Pareto improvements at both high and low learning rates and generally exhibits much less degradation.

We use the Wikipedia-grounded subset of SimpleQA (Wei et al., 2024). We evaluate on a subset of 100 questions, taking the Wikipedia documents cited for those questions and splitting them into chunks (roughly, paragraphs), resulting in a total of 1824 document chunks. As in the previous setting, we assume that the model encounters one document chunk (e.g. one paragraph) at a time and performs an immediate gradient step on the information in that chunk, rather than shuffling all chunks iid. We use Active Reading (Lin et al., 2025) to generate $N$ synthetic augmentations of the chunk. In a given batch, each sequence is a different synthetic augmentation for the same chunk. We finetune with a larger top $t = 10000$, given the higher information content in each batch.

Results are shown in Figure 4. Full finetuning and LoRA are able to perform much better than what we observed in the small-data regime as the augmented document setting is more diverse and more similar to fully iid pretraining (except for the fact that all sequences in a batch come from the same source). However, both baselines still suffer from forgetting on held-out benchmarks. In contrast, sparse memory finetuning can achieve the same target performance with much less forgetting.

## 6 ANALYSIS

**Pareto Frontier of Learning and Forgetting** There is a fundamental tradeoff between learning and forgetting: to maximize learning, one can specialize the model to a task by updating more parameters at a higher learning rate for more steps; to minimize forgetting, one can simply keep the model fixed at initialization. We plot the tradeoff in Figure 5 by sweeping across the primary parameters that control learning for each method: learning rate for full finetuning; rank, alpha, and target modules (applying lora to all linear projections or only attention weight matrices); top-$t$ and learning rate for sparse memory finetuning. We see that sparse memory finetuning indeed Pareto dominates, learning more while forgetting less.

**Naive Memory Finetuning** In Figure 6, we ablate the effects of our method by comparing to alternative ways of finetuning memory-augmented models: finetuning all accessed memory locations, and finetuning the top-$t$ indices by raw counts (i.e. TF only ranking as opposed to TF-IDF ranking).

We see using the top $t <$ all indices is enough to achieve comparable performance to finetuning all memory values, while preserving held-out performance more. If we finetune the same number of top $t$ indices with TF-only ranking, we observe comparable learning, but more forgetting. The gap between ranking with TF-IDF and ranking with TF-only widens if we finetune fewer indices ($t = 50$) on both the target and held-out tasks. This aligns with intuition: the inverse document frequency is less important if we finetune more indices (since both methods converge to finetuning all indices), but it is essential to identify the most critical indices to finetune if we restrict the number

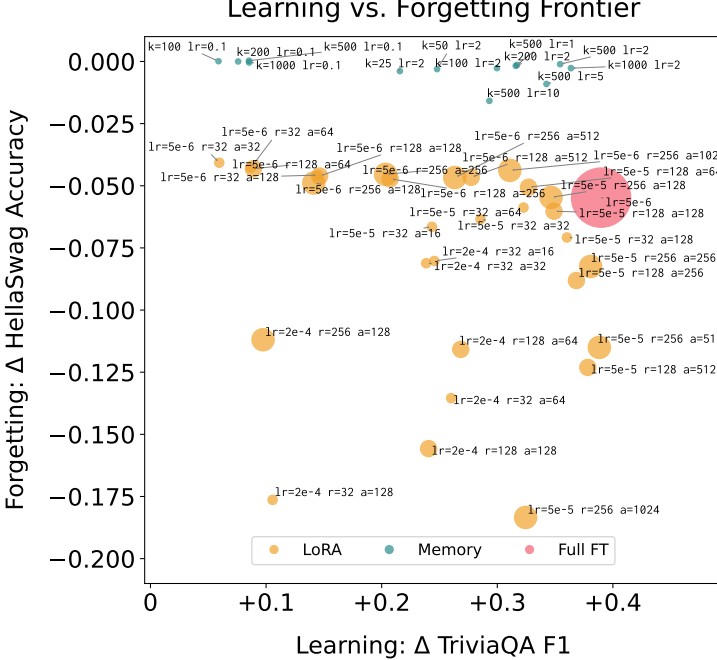

Figure 5: **Tradeoff between learning and forgetting for different methods.** To fully characterize the tradeoff for different methods (full finetuning, LoRA, and sparse memory finetuning), we plot performance on the target (TriviaQA) and a held-out task (HellaSwag) across a hyperparameter sweep. For full finetuning, we sweep across learning rate $\{2e\text{-}6, 5e\text{-}6, 2e\text{-}5, 5e\text{-}5\}$. For LoRA, we sweep across rank $\{32, 128, 256\}$, alpha $\{1/2, 1, 2, 4\}$x rank, and learning rate $\{2e\text{-}4, 5e\text{-}5, 5e\text{-}6\}$. For sparse memory finetuning, we sweep across the number of trainable indices $t$ $\{25, 50, 100, 200, 500, 1000\}$ and learning rate $\{0.1, 2\}$. We omit runs where learning fails (e.g. for full finetuning, learning rates above 5e-6 "break" the model and result in lower performance on both learning and forgetting than initialization). The size of each point depicts the number of trainable parameters per batch. We observe a continuous trend in most methods: as the learning capacity increases (e.g. with increased learning rate, or parameter count), forgetting also increases, up to a point where further increasing learning capacity leads to reduced performance. Sparse memory finetuning exhibits high learning capacity with minimal forgetting on the held-out benchmark.

of learnable parameters. TF-IDF ranking also minimizes catastrophic forgetting as some memory indices may be responsible for general token prediction (e.g. syntax or general world knowledge), and this ranking avoids "overwriting" indices that the model uses for other tasks.

**Background Indices** In Figure 7, we investigate the effect of using different background corpora to rank the trainable indices. We compare ranking against DCLM with ranking against the indices used across all TriviaQA examples (the learning set) and the indices used on Natural Questions (the held-out forgetting set). Using the indices accessed across all TriviaQA questions leads to slightly less retention and significantly more forgetting, due to the fact that we are not directly preserving knowledge learned in pretraining. Using NaturalQuestions to rank indices performs similarly to using DCLM as the background index matches the held-out evaluation. We note that the TFIDF-based ranking metric identifies indices that are commonly used across *many* NQ questions, which effectively identifies indices that are shared across the held-out "domain" (potentially explaining why performance is similar to DCLM). However, fully preserving question answering performance might require upweighting indices that are used on *any* NQ question. Future work might investigate different ways to rank that scale the access counts to weight any non-zero counts.

**Understanding Memory Accesses** To better understand how information is stored in the memory layer, we qualitatively analyze memory accesses in different sequences. Consider the TriviaQA fact learning task, where we evaluate the model's ability to answer factual questions after training

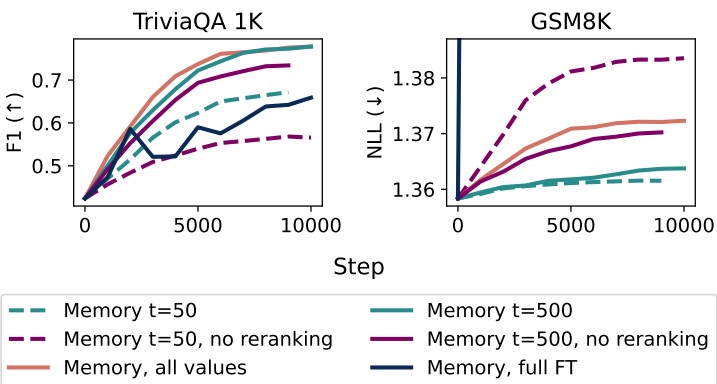

Figure 6: **Ablation: Comparison to naive memory finetuning.** We compare sparse memory fine-tuning to alternative ways to finetune memory-augmented architectures. TF-IDF ranking a subset of memory indices is sufficient to achieve similar performance as finetuning all memory values, while maintaining held-out performance. With TF-only ranking, we can retain target performance if we finetune enough indices ($t > 500$ for this task), but we observe more forgetting on held-out tasks. If we finetune fewer memory indices, the gap between TF-IDF-ranking and TF-only-ranking is more pronounced. Full finetuning the memory model (lr=5e-6) leads to much worse forgetting (outlier not shown; final GSM8K NLL is 3.87).

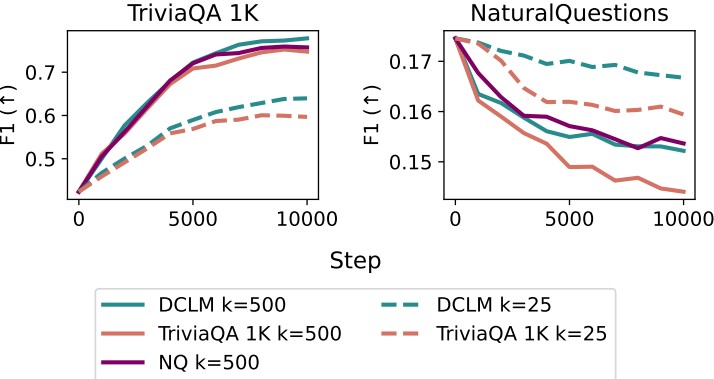

Figure 7: **Effect of background corpus.** We compare performance using different background corpora for the IDF count in the ranking score. Using the set of indices accessed on the training set itself (TriviaQA) leads to similar learning performance, but worse forgetting as we are not downweighting indices accessed on other domains where we want to preserve performance. Using the set of indices accessed on a particular set that we want to preserve performance on (NaturalQuestions) leads to similar learning and forgetting performance as using DCLM batches as a representative sample of pretraining data.

on paraphrases of each fact. In order to learn to answer questions correctly when only finetuning memory values, the training batches need to access at least one index that is also accessed in the question, and those shared indices need to be trainable. Different paraphrases access different memory indices, but we can identify the indices representing the shared "semantic" content by taking the intersection of the indices accessed on each paraphrase, and the indices accessed when answering the question. We call this set of indices the **core set**, and the size of this set gives a sense of how many indices a fact is distributed across, and thus how many we may need to finetune. We find that typical core set sizes are around 100-500 indices, confirming our intuition that parameter updates can be much more sparse than the total number of accessed indices in a batch (on the order of 1k-100k, as $(k = 32) \times (\text{num\_heads} = 4)$ gives 1024 indices accessed for each token in the batch).

When training, we don't have access to the indices that will be used during test time when answering the question. Does the TFIDF ranking procedure identify the indices in the core set, without access

Table 1: **Memory accesses at token positions on a sample of TriviaQA questions and training paraphrases.** We define the "core set" as the indices that are shared between all paraphrases and the question, and investigate whether these indices align with the top trainable indices after ranking with TFIDF. Intuitively, we would like the top trainable indices to be those that contain the "semantic" content of a factual statement that are accessed when answering the question. In the sequences below, darker colors indicate more indices at that token position are in the core (or trainable) set. Qualitatively, trainable indices tend to align with core set indices across the sequence, and often align with entity boundaries. For most questions, the minimum setting of trainable $t$ needed to answer the question correctly is much smaller than the size of the core set.

---

**Fact index: 174**   477 indices in core set, 25 indices needed to answer

**Question**

Core        How long was swimmer Michelle Smith-de Bruin banned for attempting to manipulate
            a drugs test? 4 years\<eot>

Trainable   How long was swimmer Michelle Smith-de Bruin banned for attempting to manipulate
            a drugs test? 4 years\<eot>

**Paraphrases**

Core        Michelle Smith-de Bruin was given a 4-year ban for attempting to deceive in a drugs test. \<eot>
Trainable   Michelle Smith-de Bruin was given a 4-year ban for attempting to deceive in a drugs test.\<eot>
Core        Michelle Smith-de Bruin was suspended for 4 years after attempting to deceive in a
            drugs test.\<eot>
Trainable   Michelle Smith-de Bruin was suspended for 4 years after attempting to deceive in a
            drugs test.\<eot>
Core        A 4-year ban was handed down to Michelle Smith-de Bruin for attempting to cheat on a
            drugs test.\<eot>
Trainable   A 4-year ban was handed down to Michelle Smith-de Bruin for attempting to cheat on a
            drugs test.\<eot>

---

**Fact index: 592**   169 indices in core set, 25 indices needed to answer

**Question**

Core        What was the name of the cat in Rising Damp? Vienna\<eot>
Trainable   What was the name of the cat in Rising Damp? Vienna\<eot>

**Paraphrases**

Core        A cat named Vienna appeared in the TV series Rising Damp.\<eot>
Trainable   A cat named Vienna appeared in the TV series Rising Damp.\<eot>
Core        Rising Damp features a notable feline character named Vienna.\<eot>
Trainable   Rising Damp features a notable feline character named Vienna.\<eot>
Core        The cat Vienna is a beloved part of Rising Damp.\<eot>
Trainable   The cat Vienna is a beloved part of Rising Damp.\<eot>

---

**Fact index: 83**   193 indices in core set, 100 indices needed to answer

**Question**

Core        Who was the first US-born winner of golf's British Open? Walter Hagen\<eot>
Trainable   Who was the first US-born winner of golf's British Open? Walter Hagen\<eot>

**Paraphrases**

Core        The first US-born winner of the British Open was Walter Hagen.\<eot>
Trainable   The first US-born winner of the British Open was Walter Hagen.\<eot>
Core        Walter Hagen's British Open win was a historic moment for US golfers.\<eot>
Trainable   Walter Hagen's British Open win was a historic moment for US golfers.\<eot>
Core        Walter Hagen achieved a groundbreaking victory as the first American-born winner of the
            British Open.\<eot>
Trainable   Walter Hagen achieved a groundbreaking victory as the first American-born winner of the
            British Open.\<eot>

---

to the question? In Table 1, we show the number of indices in the core set and the trainable set (for $t = 100$) accessed at each position in the sequence. Darker colors indicate more indices when predicting that token are in the core (or trainable) set. We find that the core set indices and trainable set accesses generally align across the sequence. Interestingly, we find that these indices often align with entity boundaries, hinting at where critical parametric memory reads and writes occur. Qualitatively, this suggests that TF-IDF reranking is identifying the indices that are most important for learning a fact, rather than indices that are generically useful for language modeling.

## 7  CONCLUSION

In this work, we demonstrate how sparse updates to memory layer architectures are a promising technique for continual learning without forgetting. Our key insight is that sparsity enables learning on particular inputs without interference with previously learned knowledge in the model. By leveraging sparsity inherent in memory layer architectures and selectively updating only the most relevant memory slots using TF-IDF ranking, our method enables models to learn new knowledge while keeping most parameters untouched. We find that sparse memory finetuning achieves a Pareto better tradeoff between learning and forgetting compared to full finetuning and LoRA on factual question answering tasks.

We tested our method on factual learning tasks, for which retrieval-augmented generation (RAG) is a natural present-day solution. However, our goal is to pave the way towards continual learning more broadly, enabling models to continue getting smarter over time on a broad range of tasks. For instance, we'd like language model agents to improve their coding ability as they collect more experience in the real world. It's unclear how RAG would be a suitable solution on tasks like reasoning and coding where retrieval is difficult. We'd like models to *distill* the learnings from mistakes, feedback, and reasoning chains, beyond simply storing and retrieving these episodes. As a next step, it would be important to scale our results to more complex tasks beyond fact learning, as well as larger models. Future work could also explore more sophisticated techniques for selecting the sparse set of trainable parameters, such as adapting $t$ in an input-dependent way or reranking with other criteria to push the Pareto frontier. Overall, while our work focuses on memory layers, our results demonstrate that the principle of sparse parameter updates may be a promising approach to continual learning.

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

# A  APPENDIX

# B  BASELINE RESULTS WITH SGD

| Method | | TQA 1K F1 | NQ F1 | GSM8K NLL |
|---|---|---|---|---|
| LoRA | lr=2e-5, r=$\alpha$=64 | 0.36 | 0.14 | 1.5 |
| | lr=2e-5, r=$\alpha$=128 | 0.36 | 0.14 | 1.5 |
| | lr=2e-4, r=$\alpha$=64 | 0.36 | 0.14 | 1.5 |
| | lr=2e-4, r=$\alpha$=128 | 0.36 | 0.14 | 1.5 |
| | lr=2e-2, r=$\alpha$=64 | 0.39 | 0.10 | 1.8 |
| | lr=2e-2, r=$\alpha$=128 | 0.41 | 0.11 | 1.7 |
| | lr=2e-1, r=$\alpha$=64 | 0.52 | 0.09 | 2.4 |
| | lr=2e-1, r=$\alpha$=128 | 0.58 | 0.09 | 2.2 |
| | lr=2, r=$\alpha$=128 | 0.01 | 0.00 | 10 |
| Full | lr=5e-6 | 0.36 | 0.14 | 1.5 |
| | lr=2e-5 | 0.36 | 0.14 | 1.5 |
| | lr=5e-5 | 0.36 | 0.14 | 1.5 |
| | lr=2e-3 | 0.40 | 0.11 | 1.7 |
| | lr=2e-2 | 0.56 | 0.05 | 2.0 |
| | lr=2e-1 | 0.65 | 0.01 | 3.5 |
| | lr=2 | 0.00 | 0.00 | 12 |

Table 2: Results for baseline methods with SGD for fact learning on TQA 1K. Compare to the results in Figure 3: Sparse memory finetuning with SGD achieves TQA 1K F1 > 0.7, NQ F1 < 0.15, and GSM8K NLL < 1.5. Using AdamW for the baselines learns more (TQA 1K F1 > 0.7) but forgets much more on held-out tasks, while here we see that SGD forgets less but does not learn as much.

