# OpenReview forum: "Continual Learning via Sparse Memory Finetuning"
_ICLR.cc/2026/Conference — ICLR 2026 Conference Desk Rejected Submission_

### Official Review · Reviewer_PrtR · 2025-10-31

**Soundness:** 2
**Presentation:** 3
**Contribution:** 1
**Rating:** 2
**Confidence:** 4

**Summary:**

This paper aims to address the catastrophic forgetting problem faced by LLM in continuous learning. The authors propose a sparse memory finetuning method based on a special LLM architecture with a "memory layer." The core idea is that when fine-tuning with new knowledge, instead of updating all activated parameters, the TF-IDF algorithm is used to identify and update only a small subset (top-t) of memory slots most specific to the current knowledge. This sparse update strategy aims to write new knowledge into dedicated parameters, thereby minimizing interference with the model's original capabilities.

The authors conducted experiments on fact learning and document question answering tasks, comparing their method with full fine-tuning and LoRA. Experimental results show that, while learning new knowledge, sparse memory fine-tuning significantly reduces performance degradation on retention tasks compared to baseline methods, demonstrating a superior learning-forgetting tradeoff.

**Strengths:**

1. Continuous learning and catastrophic forgetting are core challenges in deploying large-scale models, making this paper's topic relevant to real-world applications.

2. Combining the sparse activation characteristics of the memory layer with task-specific parameter updates offers some inspiration.

**Weaknesses:**

1. The novelty of this paper is relatively limited. The Memory Layer Model itself is not original; it is from existing work. The authors mainly adapt it to the specific task of catastrophic forgetting. Furthermore, the core contribution of the paper is concentrated in Chapter 4, but the TF-IDF algorithm used therein is also a mature technique. Overall, this work seems more like a combined application of existing methods, lacking significant original breakthroughs.

2. The paper mentions experiments on a 1.3B-scale model, but does not specify which type of model architecture was used. More importantly, all experiments were conducted on a relatively small-scale model, making it difficult to fully verify the scalability and generalization ability of the proposed method on larger models. In addition, the memory layer model requires replacing the FFN in the original model with a memory layer structure, which may bring additional optimization challenges in large-scale models—the authors do not discuss whether the performance of downstream tasks can still be maintained on larger models.

3. The experiments were only conducted on two general QA datasets, resulting in a relatively limited evaluation scenario. To fully validate the method's generality, the authors should supplement it with more domain-specific fine-tuning experiments (such as in medical and legal fields) and adopt more diverse evaluation benchmarks to more comprehensively measure the method's effectiveness in mitigating catastrophic forgetting.

4. The paper only reports the absolute performance of the proposed method but does not provide a systematic comparison with current mainstream catastrophic forgetting mitigation methods. Therefore, readers cannot determine the method's actual competitiveness and advantages among existing technologies.

**Questions:**

see weakness

---

### Official Review · Reviewer_LZhp · 2025-10-31

**Soundness:** 1
**Presentation:** 3
**Contribution:** 2
**Rating:** 2
**Confidence:** 4

**Summary:**

The authors tackles the problem of continual learning and propose Sparse Memory Finetuning (SMF). SMF fine-tunes a memory layer that replaces the transformer FFN with product-key lookup matrices. During fine-tuning, the method counts which memory indices are accessed by a batch, ranks indices via TF-IDF relative to some background corpus, and updates only the top $t$ value vectors while freezing everything else.
During inference, the model just performs the sparse memory lookup top-$k$ retrieval per memory head. The authors consider two cases: 1) streaming fact learning and 2) streaming document learning. The paper shows that SMF has similar target-task learning to full FT and LoRA but dramatically less forgetting on held-out tasks (NQ / HellaSwag / GSM8K). The authors analyzed TF-IDF vs TF-only ranking and the choice of background corpus.

**Strengths:**

- The core idea is pretty easy to follow
- The idea is clear, simple, and is easy for the community to build upon
- Conditioning on a background corpus is a novel approach to control what kind of forgetting one is targeting to prevent; this might be an interesting new design choice to explore
- I think the evaluation setup makes sense to test the claim (e.g., the fact vs doc learning settings) with caveats described in the following section

**Weaknesses:**

Although the main idea is appealing and the experiments show positive signals, the coverage of the experiments seems rather limited.

**Major issues**
- One main result is missing (i.e., the results for Document QA). This is the main reason for the low soundness rating.
- Often times the benefit of forgetting mitigation simply comes from the reduced number of parameters that are updated. The flip side is the reduced learning capacity. While some of this is tested in the paper (Fig 4), the experiments don't seem comprehensive enough to show that the learning capacity of the memory layer is as good as LoRA or Full FT only base on the TriviaQA learning result. One experiment to test is multi-stage fine-tuning and see if sparse update actually has less forgetting throughout.

**Analyses that i think would strengthen the argument of the paper**
- Along the line of memory layer capacity, it would be great to see the a sweep top $k$ indices in the memory layer is fine-tuned.
- The background corpus analyses in the paper are valuable, but I would love to see a different corpus tested beyond DCLM. Especially, if one wants to preserve the capability of a particular domain, does choosing a corresponding background corpus help this?
- "We note that ranking based on batches makes no assumption about task boundaries; consecutive batches can be from the same or totally different data distributions." It would be really nice to see this tested, which can be done with the aforementioned sequential training setting.

**Minor**
- L40: a citation is missing


While I appreciate the idea, my general sense is that the experiments and analyses are a bit limited or might be over-indexing on the specific datasets. I would be happy to increase my scores if the above points are addressed.

**Questions:**

- Why only insert one memory layer? What would happen if the memory layer is inserted throughout the model?
- Why is GSM8K measured in NLL and not the standard accuracy? It doesn't seem to be justified in the paper.
- Fig 6: should the caption be $t$ instead of $k$?

---

> ### Author Response · Authors · 2025-11-20
> **Draft with SimpleQA experiments**
>
> We're sorry about the erroneous draft! Regarding your major concerns, could you take a look at the revision and let us know if they are addressed?
>
> Specifically:
> > One main result is missing (i.e., the results for Document QA). This is the main reason for the low soundness rating.
>
> Please see Figure 4.
>
> > Along the line of memory layer capacity, it would be great to see the a sweep top $k$ indices in the memory layer is fine-tuned.
>
> To clarify, does the sweep across number of finetunable memory indices in Figure 5 (Pareto plot) already address this question?

---

### Official Review · Reviewer_fD2i · 2025-11-01

**Soundness:** 3
**Presentation:** 3
**Contribution:** 2
**Rating:** 4
**Confidence:** 4

**Summary:**

This paper addresses catastrophic forgetting in continual learning for large language models by proposing Sparse Memory Finetuning, a method that updates only a small number of memory slots that are highly activated by new data but rarely used during pretraining. Experiments show that this method learns new knowledge effectively while greatly reducing forgetting compared to full finetuning and low-rank adaptation.

**Strengths:**

- This approach does not require any architectural changes beyond adding memory layers.
- The sparse update mechanism limits parameter interference, preserving previously learned knowledge and mitigate the forgetting.
- This method enables continual learning without requiring data replay, which improve the scalability and efficiency.

**Weaknesses:**

- Evaluation is limited to two QA tasks (fact learning and document QA), it would be great to have the result in multi-task, reasoning, or multilingual continual learning.
- Experiments only include full finetuning and LoRA, I think the authors need to compare with other continual learning approach as the baseline. For now, it unclear how the method compares to those.

**Questions:**

- If the new knowledge is similar to previously learned knowledge but not identical, will the model choose to update entirely new memory slots, or will it incorporate and refine the existing knowledge stored in the previously used slots?
- You evaluated your method only on a 1.3B parameter model. Could you clarify why this particular scale was chosen, and whether you expect the results to hold for larger models? Usually, we use smaller-scale model for interpretability, but I did not see explicit interpretive analysis in the paper.

---

### Note · Program_Chairs · 2026-01-17
**Submission Desk Rejected by Program Chairs**

The following references in this submission do not refer to real documents and/or have major errors in bibliographic information:

 Thomas Scialom, Paul-Alexis Dray, Sylvain Lamprier, Benjamin Piwowarski, and Jacopo Staiano. Continual learning for large language models: Improving zero-shot generalization via memory replay. In NeurIPS Workshop on Efficient Continual Learning, 2022.